# Framework of Virtual Plantation Forest Modeling and Data Analysis for Digital Twin

**Wanlu Li** [1,2]**, Meng Yang** [1,2,]*****, Benye Xi** [3] **and Qingqing Huang** [4]

1 School of Information Science and Technology, Beijing Forestry University (BFU), Qinghua East Road 35, Haidian District, Beijing 100083, China
2 Engineering Research Center for Forestry-Oriented Intelligent Information Processing of National Forestry and Grassland Administration, Beijing 100083, China
3 Ministry of Education Key Laboratory of Silviculture and Conservation, Beijing Forestry University (BFU), Qinghua East Road 35, Haidian District, Beijing 100083, China
4 School of Technology, Beijing Forestry University (BFU), Qinghua East Road 35, Haidian District, Beijing 100083, China
* Correspondence: yangmeng@bjfu.edu.cn

**Abstract:** Plantation forests, cultivated through artificial seeding and planting methods, are of great significance to human society. However, most experimental sites for these forests are located in remote areas. Therefore, in-depth studies on remote forest management and off-site experiments can better meet the experimental and management needs of researchers. Based on an experimental plantation forest of *Triploid Populus Tomentosa*, this paper proposes a digital twin architecture for a virtual poplar plantation forest system. The framework includes the modeling of virtual plantation and data analysis. Regarding this system architecture, this paper theoretically analyzes the three main entities of the physical world, digital world, and researchers contained in it, as well as their interaction mechanisms. For virtual plantation modeling, a tree modeling method based on LiDAR point cloud data was adopted. The transitional particle flow method was proposed to combine with AdTree method for tree construction, followed by integration with other models and optimization. For plantation data analysis, a database based on forest monitoring data was established. Tree growth equations were derived by fitting the tree diameter at breast height data, which were then used to predict and simulate trends in diameter-related data that are difficult to measure. The experimental result shows that a preliminary digital twin-oriented poplar plantation system can be constructed based on the proposed framework. The system consists of 2160 trees and simulations of 10 types of monitored or predicted data, which provides a new practical basis for the application of digital twin technology in the forestry field. The optimized tree model consumes over 67% less memory, while the $R^2$ of the tree growth equation with more than 100 data items could reach more than 87%, which greatly improves the performance and accuracy of the system. Thus, utilizing forestry information networking and digitization to support plantation forest experimentation and management contributes to advancing the digital transformation of forestry and the realization of a smart management model for forests.

**Keywords:** digital twin; forest modeling; virtual reality; forest management; data analysis

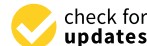



## 1. Introduction

The role of plantation forests is crucial in achieving sustainable development and mitigating the impacts of climate change [1]. Plantation forests are cultivated by artificial seeding, planting or cutting methods. They have clear operational purposes and are an important component of forest resources in reducing atmospheric $CO_2$ levels [2]. Several countries have developed plantation forests for many different functional purposes, such as ecological public interest forests and industrial timber forests, which have made important contributions to ecological restoration [3], environmental improvement [4], and timber

security [5]. Due to the special characteristics of experimental space, plantation forests are often located in remote areas, occupying a large area with numerous trees. Therefore, it is challenging to allocate personnel for plantation forest cultivation and management, which requires a considerable amount of manpower, material, and effort. Moreover, the data obtained through traditional measurement methods may have certain abstraction, and the completeness and accuracy may also be affected. In the current rapidly developing information technology environment, it is essential to integrate plantation forests with emerging technologies, and the development of intelligent forest management [6–8] becomes an important issue worth considering and discussing.

Currently, the most widely used field for forest management applications is forest fire prevention [9], which includes some intelligent applications, such as detecting flames using lightweight convolutional neural networks [10] and detecting early smoke from fires based on support vector machine (SVM) image segmentation [11]. In addition, there are other intelligent applications in forestry, such as using machine learning to evaluate the suitability of plantation sites [12] and using neural networks to enhance height predictions of mixed forests with uneven age [13]. However, there are still problems in the construction of forestry informationization, such as the need for improved forestry databases [14], lagging forestry information infrastructure [15], and low cohesion between forestry work and information technology [16]. At the same time, the limitations of location, time, and field observations have not been eliminated in forestry research, and it is still difficult to remotely monitor, control, and coordinate forest operations. Therefore, combining digital twin technology for virtual forest modeling and data analysis can effectively help forest management move towards digitalization and informationization.

Digital twin technology [17] is a popular current research area that combines artificial intelligence, sensors, and virtual reality to map the full life-cycle processes of physical equipment by generating virtual models in a digital space. This concept was first applied in aerospace [18], and it can support simulation-related applications such as analysis and prediction and feed the results back to the physical object to help in optimization and decision making [19]. Digital twin is a widely applicable theoretical technology system that can be applied to product design [20,21], engineering construction [22,23], water conservancy and hydropower [24,25], and biomedicine [26,27]. Among them, there are richer applications in smart cities [28,29], smart industries [30–32], and smart agriculture [33,34]. With the help of digital twin technology, research in forestry can achieve a transition from traditional empirical methods to precise scientific methods. Digital twin technology can not only establish high-precision models of trees and optimize forest management strategies, but also help researchers better understand the operational mechanisms of forest systems, thus improving production efficiency, reducing production costs and promoting sustainable development. However, the application of digital twin technology to forestry is still a huge challenge.

As a unique entity, the forest is characterized by complex and diverse data, difficulties in 3D modeling, and challenges in technical implementation. To address these issues, many researchers have been exploring new methods and technologies. For simulating forests, individual plants can be modeled based on their living environment and botanical knowledge [35], or partial tree modeling can be achieved through imaging [36] and other means. Markus et al. [37] classified, detected, and reconstructed individual canopies of urban forests by point clouds, and reduced data gaps by parametric reconstruction of canopies. Guo et al. [38] proposed a new method for constructing realistic sense plant models. Jiang et al. [39] proposed a digital twin method based on long short-term memory (LSTM) to model forests using satellite remote sensing images. Light Detection and Ranging (LiDAR) [40] is also an advanced active remote sensing technology that can quickly and accurately acquire spatial 3D coordinate information over large areas. Daniel [41] used GeoSLAM mobile LiDAR scanner and Virt Silv AI platform to examine the individual tree segmentation technique efficiency, providing higher accuracy and precision for 3D trees in digital twin. Du et al. [42] proposed the AdTree approach to 3D reconstruction of

trees under laser scanning, with an overall fitting error of less than 3 cm. It appears that currently, most applications for modeling or managing forest objects are based on one-sided virtual reality (VR) or Internet of Things (IoT) technologies, with fewer comprehensive forest digital twin applications and research on visualizing forest management.

Based on the work in the experimental forest of *Triploid Populus Tomentosa* in Qingping, Shandong, this study has constructed a framework for virtual plantation forest modeling and data analysis oriented towards digital twin. The main contributions are: (1) proposing a digital twin-based virtual poplar plantation forest system architecture for forest management structures and systematically analyzing the main role of the architecture mechanism; (2) based on collected point cloud data from the forest, proposing a transition particle flow method combined with the AdTree method to model trees and establishing and optimizing virtual plantation forest scene on this basis; and (3) based on the experimental monitoring data, analyzing forest monitoring data and constructing a database, fitting tree growth formulas based on the measured tree diameter at breast height data, and thus predicting and simulating related physiological indicators. From this, a virtual poplar plantation system can be initially built based on virtual reality to support the realization of the virtual forest management model through the networked and digital application of forest land information.

## 2. Methods

The plantation forest in Qingping, Shandong Province, is a relatively gentle plain terrain with one *Triploid Populus Tomentosa* sapling planted at 2 m longitudinal and 3 m lateral intervals. A total of 2160 poplar trees seedlings were planted in this stand area.

### 2.1. Framework Overview

According to the basic experiments and overall requirements of poplar plantation, this paper proposes a digital twin-based virtual poplar plantation system as shown in Figure 1, which consists of three main bodies: the physical world, the digital world and the researchers.

The physical world is the real experimental forest, which includes the actual forest management, forest operation, and various sensing and monitoring equipment; the digital world is a digital "twin" of the physical world, which can be primarily divided into virtual forest component, data analysis component, object interaction component, etc., based on its functional requirements; and the researchers are the actual operators of the architecture. In the perfect poplar plantation forest digital twin system, while the physical world conducts its own management, experiments and operations, the real information and data captured and collected, such as pictures and videos, are transmitted to the digital world in real time. The digital world focuses on the elements of forest ecological environment and employs integrated analysis and mixed modeling to decipher its complexity. Upon receiving information and data from the physical world, the digital world processes and performs intelligent calculations. After simulation and deduction, the digital world provides feedback and decisions to researchers and the physical world to support decision-making and optimization for forest experiments and management. In response to the feedback, the physical world implements intelligent instructions to maintain or promptly change controllable variables of the forest, thereby promoting the conduct of physical world experiments. Meanwhile, researchers collect feedback from both the physical and digital worlds and make corresponding adjustments, thus forming a large digital twin cycle.

Currently, the overall management and experiments of the physical world and digital world in the poplar plantation are operated by "researchers". Due to the limitations of the experimental site, technology and equipment, the current physical world has not been able to achieve adaptive intelligent instruction control of variables. In addition, the digital world currently does not consider the complex effects among various factors, and direct information feedback to the physical world can only be achieved through manual intervention. On this basis, this paper mainly aims to discuss the framework of

virtual forest modeling component and data analysis component in the digital world of the poplar plantation.

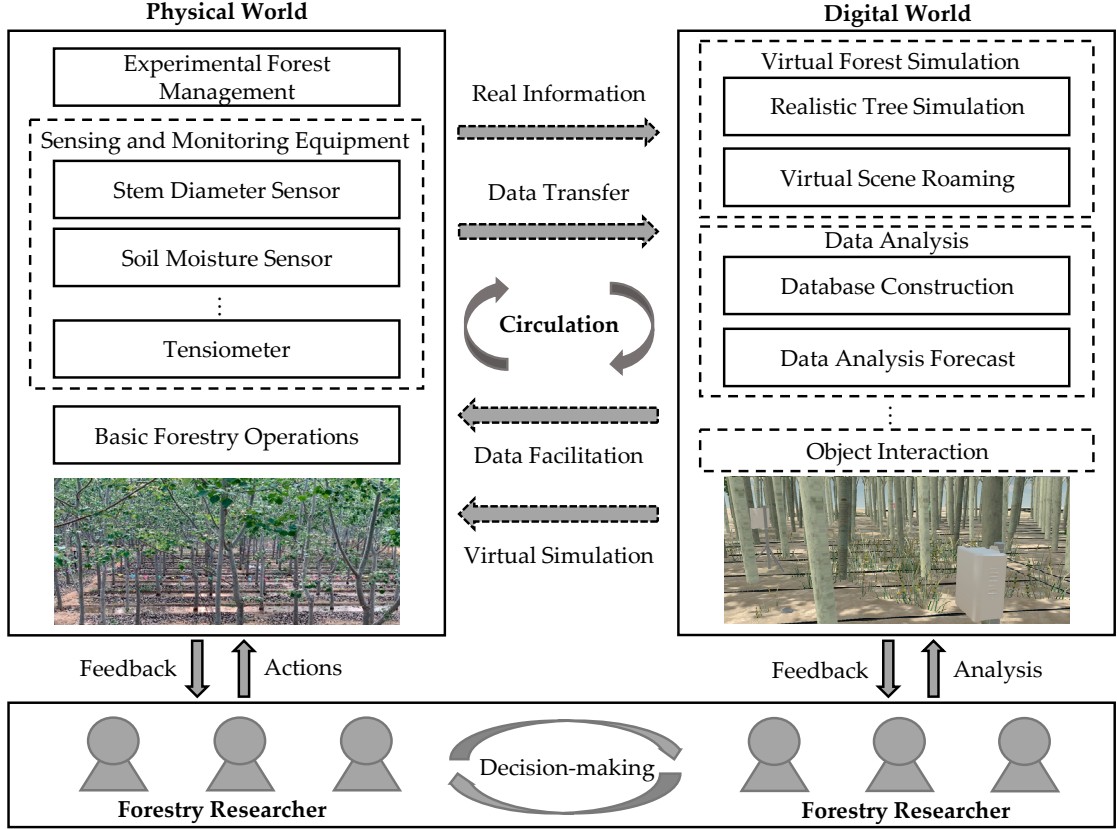

**Figure 1.** Architecture of virtual poplar plantation system based on digital twin.

In the system architecture of the poplar plantation, the virtual forest of the digital world is the basic carrier of the whole system. It needs to simulate the main research objects and build a virtual plantation scene. The application data in the virtual forest mainly consists of images and point cloud information from the physical world. In terms of data analysis, its main task is to connect and present data from the physical world, including the display of sensor monitoring equipment data and the analysis and prediction of some data. The main input data for data analysis are sensor monitoring equipment data and some manually measured data collected periodically. In this way, key issues analysis and research of the components and functional implementations are carried out, laying a certain foundation for the future construction of the virtual forest digital twin system.

## 2.2. Virtual Forest Modeling

This section mainly introduces the relevant methods for constructing virtual forests. The specific flowchart is shown in Figure 2.

### 2.2.1. Point Cloud Data Pre-Processing

The airborne LiDAR scanning system can acquire surface tree point cloud data on a large scale, and it can contribute to high-precision tree structure parameter extraction and landscape level geometric reconstruction. The raw point cloud data is large (Figure 3a), and there are many trees and intricate branches in the woodland, so the completeness of the edge point cloud is not considered first. An area of the stand point cloud was cut out to segment the trees using the method of Zhang et al. [43]. With the high central position of the tree canopy and the low surroundings, as well as the gradually increasing height from the surroundings to the center, the highest point of the tree point cloud can be projected onto

the ground plane, which will be distinguished to form a height map and thus distinguish the single tree point cloud.

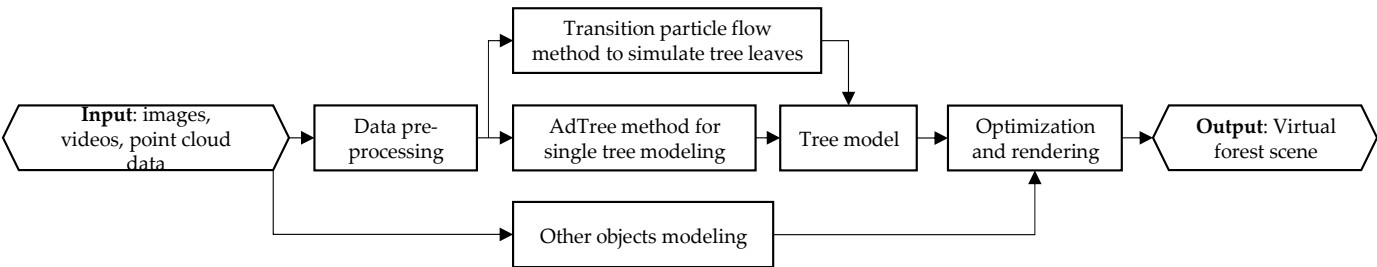

**Figure 2.** Flowchart of virtual forest modeling.

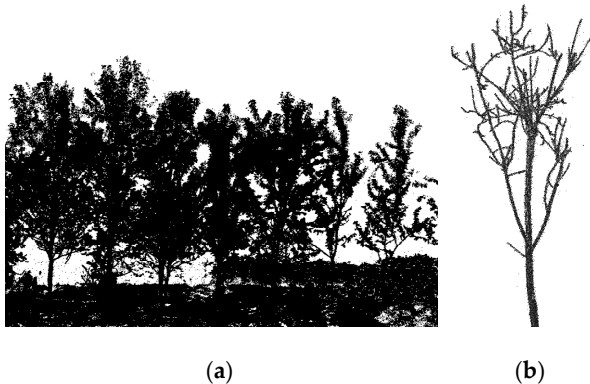

(**a**)　　　　　　　　　　(**b**)

**Figure 3.** Forest site point cloud data. (**a**) Local point cloud data of poplar plantation. (**b**) Point cloud data of tree branches and trunks.

Noise in point cloud data can be divided into two types: large-scale noise and small-scale noise. Large-scale noise refers to the point cloud data of the ground that is scanned over a large area, while small-scale noise refers to outliers around trees. In order to filter out these outliers, a pass-through filtering method can be used. Generally the point cloud density of the branch part of the tree will be greater than the density of the leaf point cloud. Therefore, in the point cloud data, the lowest point in the normal direction is proposed as the initial root node, and the radius is set. The kd-tree [44] method is used for search traversal, and points with a distance less than the radius to the root node are continuously added to the branch point cloud set. Points outside the radius distance are counted as the leaf point cloud set. The leaf point cloud set is removed to obtain the branch and trunk point cloud data (Figure 3b).

### 2.2.2. Tree Modeling

In this paper, the AdTree method is mainly used for the trunk reconstruction of poplar trees. For tree trunk point cloud data, the main idea is to construct a tree branch model by getting the initial tree skeleton to fit the columns. The Delaynay method can generate a set of triangles, or triangular mesh, for a given set P of planar points. However, the generated triangle mesh is not unique, and the choice of triangle edge length vertices determines the type of adjacent triangles (Figure 4). The empty circle property of Delaynay is satisfied when all the external circles of the triangles do not contain any vertices in P in the range other than the boundary (Figure 4a). In the tree point cloud data, the points with similar intervals are likely to belong to the same branch. The point cloud data after performing Delaynay processing can find its minimum spanning tree MST to obtain the initial skeleton

information of the tree model. By assigning weights to the edge lengths in space through the Euclidean metric, the weights of the edge lengths in space, *De*, can be expressed as:

$$De = \sqrt{\sum_{i=1}^{n}(x_i - y_i)^2} \qquad n = 1, 2, \ldots, n \tag{1}$$

where the coordinates of the vertex *n* are $(x_n, y_n)$.

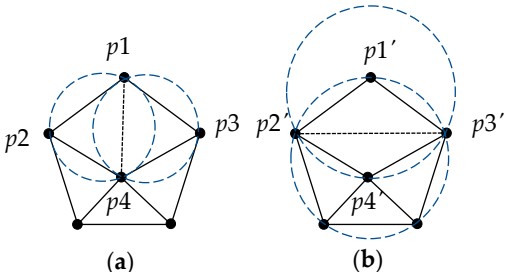

(a) (b)

**Figure 4.** Delaynay triangulation with the empty circle property. (**a**) The outer circle of a triangle formed by *p*1, *p*2 and *p*4 contains only the points of the triangle. (**b**) The outer circle of the triangle formed by *p*1′, *p*2′ and *p*3′ contains *p*4′ and other vertices.

Then the Dijkstra algorithm is used to search the shortest path of the point cloud with breadth first. The point set S is used to store the vertices on the path, and the point set U is used to store the non-path vertices, {S,U} ∈ P. We defined the array dis to store the weights from the origin to each vertex. We iterated through the weights from the origin to the vertices, added the vertices to the point set S when the weights were smallest, and updated dis. We kept cycling through this process until all the vertices were included in S, thus obtaining the initial skeleton representation of the tree MST.

For a branch of a tree, the general simulation method is to use cylinders or polygons to represent it. If the actual radius value is known, a series of cylinders can be fitted based on the tree skeleton to accurately simulate the tree branch. The point cloud data of the tree trunk and main branches is relatively dense and stable, and accurate branch radius can be obtained using non-linear least squares method. The point cloud data near the tree crown and the top of the branches becomes noisy and obtaining accurate radius data is difficult. In this case, the correspondence between the radius of the tree trunk and the radius of the sub-branches can be expressed as:

$$R^2 = \alpha \sum r_n^2 \qquad n = 1, 2, \ldots, n \tag{2}$$

where *R* is the fitted radius of the main branch, *r* is the radius of the sub-branch, and *α* is the ratio of the thickness of the main branch to the sum of the thicknesses of the side branches.

It is worth noting that the input point cloud data still contains some redundant and noisy points that have not been removed during preprocessing. These points are represented as redundant and meaningless branches in the skeleton, which exacerbate the effects of overlapping and invalid operations during fitting, and the resulting tree model will appear complex and rough. Therefore, before fitting, it is necessary to simplify the tree skeleton. In AdTree, by assigning different weights to vertices and edges and iteratively checking the proximity between adjacent points, some vertices are merged to simplify the tree skeleton, as shown in Figure 5. This approach can not only eliminate the impact of skeleton branching at noise points, but also make the overall skeleton more concise and intuitive. In the actual generation process, direct cylindrical fitting based on the tree skeleton sometimes appears too rigid at the turning and intersection points, and the sharp transition will lead to the loss of realism of the tree model. Therefore, the tree skeleton can be curved and optimized before fitting to make the model more natural and smooth.

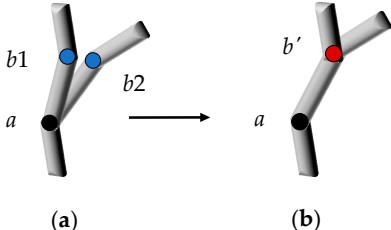

**Figure 5.** The similarity between vertices is checked according to different weights. (**a**) Points $b1$ and $b2$ have high similarity. (**b**) Point $b'$ is merged from $b1$ to $b2$, thus simplifying the tree skeleton.

The next step is the simulation of tree leaves. Due to the overlapping part of the point cloud data information of the tree leaves, it is relatively difficult to reconstruct the tree leaves by means of the point cloud information. Therefore, a transition particle flow method is proposed to simulate the leaves of poplar trees. As shown in Figure 6, the completed trunk model is first labeled. Then, the normal vertex O of the model is adjusted to be close to the ground, which is the lowest point in the center of the model. We set the height $h$ upward along the normal direction with O as the demarcation of the particle emission point of the tree leaves. $h'$ is the transition value of the emitted particle stream. There exists a point $a(x_a, y_a)$ at the partition, upward along the normal direction, at a distance $h'$ to obtain the point $a'$, $h'$ can be expressed as:

$$h' = |y_a - y_{a'}| \tag{3}$$

where $y_a$ takes a value equal to $h$. We performed the calculation of the downward point $a''$ similarly. The area with a height of $2h'$ is the particle flow transition zone, and the part above the transition zone is the random particle flow zone, and the part below is the empty zone. Generally, the value of $h'$ is set between the main trunk and multiple branches, thus simulating the characteristics of real trees with few leaves at branches.

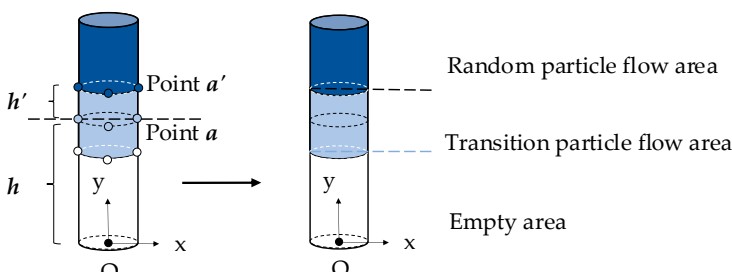

**Figure 6.** Particle flow emission region delineation. O is the coordinate origin. A dividing line is made through point $a$, which is parallel to the x-axis at a distance $h$.

For the empty area, part of the vertices are marked with the color white, the random particle flow area vertices are marked with black, and the transition particle flow area takes two color transition values. It is worth noting that the number of leaf particles in the random particle flow area should be much more than the number of transition areas. Then the number of particles in each region is expressed as:

$$F_{\text{leaf}} = N_{\text{tree}}(k \times RGB_{\text{black}})F \tag{4}$$

where $F_{\text{leaf}}$ is the actual number of particles in the region, $F$ is the initial number of emitted particles, $RGB_{\text{black}}$ is the region blackness value, $N_{\text{tree}}$ is the number of tree model vertices in the region, and $k$ is the percentage of particles emitted under $RGB_{\text{black}}$. When $k = 80\%$ and $RGB_{\text{black}} = 50\%$ in the transition particle flow region, a more ideal state of the tree leaves emitting particles can be achieved. The particle emitting operation is performed on the overall trunk model to detect the vertex color marker, and the vertices marked as white

will not show the emitting particles. The transition area has a higher value of whiteness, but still shows some particles. We set the emitted particles as leaf-shaped facets and the material as a poplar tree leaf map. We randomized the particle size and rotation angle, and emitted according to the markers to obtain the leaf particle stream. The higher the number of particles initially emitted, the higher the overall number of leaves generated. By controlling the amount of foliage to select the season and time of year of the aspen trees, the virtual scene is more selective.

### 2.2.3. Virtual Plantation Simulation

The objects of the virtual plantation forest mainly contain poplar trees, woodland topography and monitoring equipment. The tree model is obtained by integrating the trunk model and leaf particles based on Section 2.2.2. Then, the model height values are assigned within a certain fluctuation range interval based on the tree height data obtained from different processing methods according to the tree growth formula in Section 2.3.3. For the plantation forest terrain, the Terrain component of Unity3D is used to complete the drawing of flat terrain and artificial border irrigation. In addition, to simulate the distribution of trees according to the real layout, we decorated with weeds and wildflowers and other models to increase the sense of realism. It can also used to animate water materials to fill the gullies, add sky boxes to set the overall environmental tone, set parallel light sources, point light sources to simulate natural sunlight and environmental reflections, adjust the scene shadows, and set the wind field and other environmental details to build a high degree of completion of the virtual forest scene. The monitoring equipment mainly includes equipment for forest management, experiments and instruments on tree physiological indicators, such as weather stations, groundwater level sensors, etc. Based on the real pictures and videos of the plantation forest and other information, 3dsMax was used to assist in modeling it.

### 2.2.4. Scene Optimization

The virtual forest contains a large number of models with a high degree of redundancy. There are different sizes and levels of detail depending on how much detail is required. The simulation of the environment will also involve some specific dynamic scenes, which will inevitably lead to serious memory consumption, rendering pressure, and occasional lagging problems.

The main model in this paper is a poplar tree model, which has the problem of cluttered lines and faces in wiring and arranging points, and thus, it is very laggy when loading and rendering the tree model continuously. In order to improve the operation efficiency, we rewired the tree model and performed operations such as model attachment to reduce the number of points and lines. The Levels of Detail (LOD) [45] technique was used to simplify the level of detail in the complex model during rendering. In addition, dynamic loading techniques were used to reduce the burden of running the system scene. In the process of following the camera tour, the camera viewpoint range is limited, and the scene models within the viewpoint range are continuously loaded while the models outside the range are continuously deleted to save the memory of the scene operation. Reasonable use of buffer pools for caching prefabricated objects and finding useless and duplicate content in project resources for early deletion can also better reduce the storage pressure on the system.

In the framework of the digital twin system, the virtual poplar plantation model is mapped to the real forest. The real forest transmits information to the virtual forest to simulate the real effect, which causes the problem of model updating. For the update of the plantation model data, the long growth cycle of *Triploid Populus Tomentosa* in poplar plantation, which has to be monitored for a long period of time on an annual basis before conclusions can be drawn, makes it difficult to observe direct stand changes in a relatively short period of time. The specific updates of the objects in the stand for the current practical

research needs are shown in Table 1. Further possibilities need to be explored for a more efficient update method.

**Table 1.** Renewal of virtual poplar plantation forest objects.

| Plantation Forest Object | Update Status | Update Cycle |
|---|---|---|
| Poplar tree model | Need to be updated | 4 Months |
| Plantation forest terrain | No need to update | 0 |
| Models of monitoring equipment already in existence | No need to update | 0 |
| Newly added monitoring equipment models [1] | Need to be updated | 4 Months |
| Other models (weeds, etc.) | Need to be updated | 4 Months |

[1] These devices may be added to real plantation forests in the future.

### 2.3. Data Analysis

By using tree growth equations and existing data, it is possible to predict the biomass of trees that are difficult to measure. Predictions involve assessing the future growth status of trees, and accurate predictions can facilitate guidance for field experiments. For future monitoring data, exponential smoothing algorithms can be used to simulate the past and current data, and the goodness of fit $R^2$ can represent its prediction quality. Thus, by combining existing breast diameter data with data visualization methods, the trends in tree growth can be intuitively represented in the system, and the future trends and range of the data can also be clearly indicated.

Analyzing and predicting the data can provide more effective decision-making for experimental forest management and basic field work. In combination with the experimental results of researchers, corresponding solutions can be proposed for abnormal situations that may occur in the analysis of forest data, such as excessively high peaks. Targeted measures can be formed and feedback can be provided in a timely manner when such situations arise. This can help forestry researchers prevent events such as forest droughts and fires.

### 2.3.1. Database Construction

A SQL Server database is created to store the monitoring data, mainly in the form of tables. Most of the data are automatically collected by sensors set up in the forest and then collected by data collectors. A small amount of the data is manually collected, processed and uploaded to the database. Due to the large amount of redundant and complex content in the data collected by the data collector, preprocessing is required. Empty or invalid semantic fields are queried and deleted, and the necessary fields are retained and sorted. However, since the database serves as a carrier for data storage, it is difficult to analyze or use it intuitively. Therefore, it is necessary to consider connecting the database to the system for further analysis and use, as shown in Figure 7.

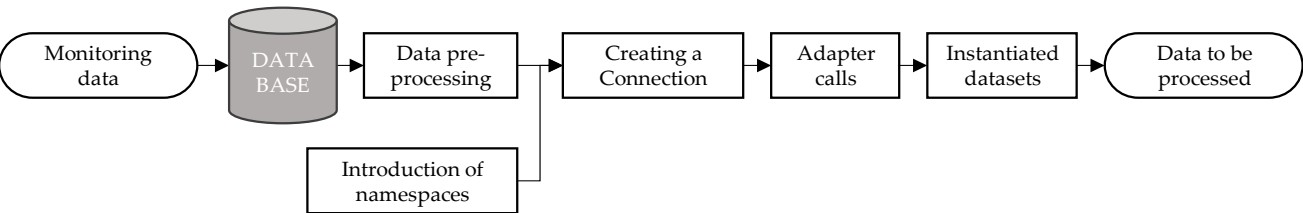

**Figure 7.** Flowchart of building a database and connection.

Add the relevant class libraries to connect to the SQL Server database in the system project. In the NET framework, the System.Data.SqlClient namespace is introduced to create objects that receive data variables and connect to the database through the SqlConnection object. Connection is equivalent to the connection channel between the system and the database, and the main access type is determined by the database. Command can execute commands and return results from the source data. We used SqlDataAdapter adapter

to make database table calls and create DataSet for instantiation. DataAdapter acts as a connection between the instantiated object and the source data, it can fill the DataSet by retrieving the source data, and also modify the source data by Update. After instantiating the data into the DataSet object, it can directly call the data for analysis and simulation, etc., and also conduct the foundation for future data linkage in real time.

### 2.3.2. Data Visualization

Some of the continuous data from the DataSet were selected for visualization. In this paper, three parts of sensor monitoring data are selected for visualization and analysis: groundwater depth data, tension gauge data and stem diameter sensor data. Their main semantic fields are date, different processing methods of experiments, and different monitoring points. Combining the characteristics of the data and the needs of foresters, line charts is used to present the trend changes between the different data. Using the date of data monitoring as the x-axis and the data of different treatments as the y-axis, multiple line graphs are drawn by UGUI controls. If the data from the same device includes multiple monitoring points, they are marked using different legends and colors to facilitate the observation of status information of different tree collection points in the same time state.

### 2.3.3. Specific Data Analysis and Simulation

For stand growth status, predicting tree growth under forest management status from specific data or simulating future trends of impact factors can help researchers to develop subsequent management plans and disaster prevention. In this paper, we choose to simulate tree height, above-ground biomass, below-ground biomass, and total biomass as physiological indicators through regular manual measurements of tree diameter at breast height (DBH).

The measurement of tree DBH is relatively easy compared to the measurement of the physiological indicators that are to be predicted for the simulation. Based on the knowledge and laws of forestry, the existing tree DBH and tree height data were selected for scaling, and a proportional logarithmic relationship was found based on the scatter direction. From this, it is presumed that a function fit can be performed to obtain the tree growth equation about tree height. Other physiological indicators were also fitted by the same method, and some of the scatter plot results were obtained as shown in Figure 8.

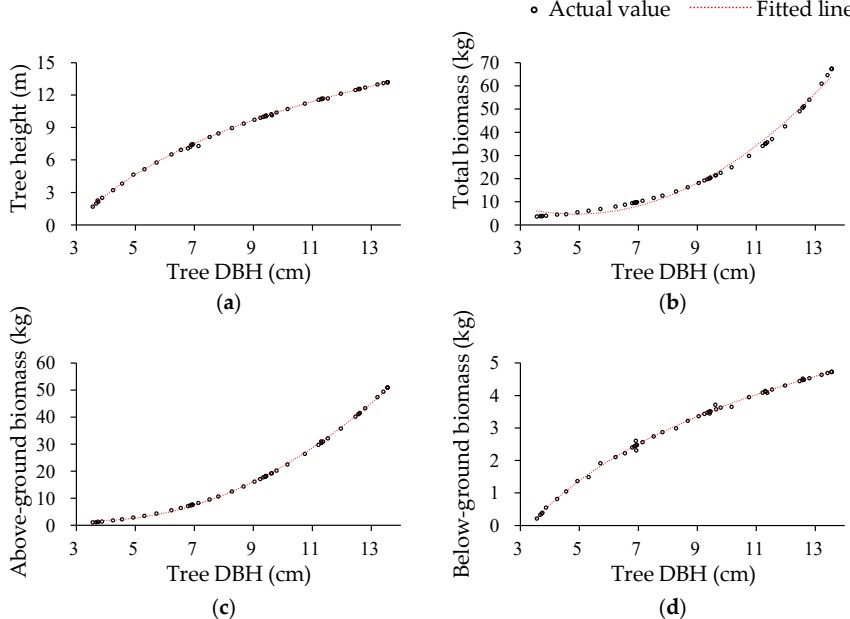

**Figure 8.** Under adequate drip irrigation treatment, scatter plots with fitted curves were constructed for some tree diameter at breast height (DBH) data as follows: (**a**) the fitted curve of DBH and tree

height data; (**b**) the fitted curve of DBH and total biomass (excluding leaves and defoliation); (**c**) the fitted curve of DBH and above-ground biomass (including only trunk and branches); (**d**) the fitted curve of DBH and below-ground biomass.

In general, the fitted tree growth equation can assist in determining future physiological indicators and reduce errors to some extent. Combining different tree treatments in plantation (adequate drip irrigation, water-controlled drip irrigation, adequate furrow irrigation, water-controlled furrow irrigation), within a certain range of DBH, for tree height there is a relationship as shown in Equation (5):

$$H = \alpha_1 \ln D - \beta_1, \; 3.47 < D < 16.2 \tag{5}$$

where $H$ is the tree height, $D$ is the tree DBH, $\alpha_1$ takes the value of 8.586 and $\beta_1$ takes the value of 9.1749. The relationship between DBH and total biomass $B_S$ for a given range is shown in Equation (6):

$$\begin{cases} B_{S1} = \alpha_2 e^{\beta_2 D} \\ B_{S2} = \alpha_3 e^{\beta_3 D} \end{cases}, \; 4.08 < D < 16.02 \tag{6}$$

where $B_{S1}$ includes trunk, branch, root stump, thick root and thin root parts, $\alpha_2$ takes the value of 1.3121 and $\beta_2$ takes the value of 0.2907; $B_{S2}$ increased both leaves and deciduous leaves compared to $B_{S1}$, with $\alpha_3$ taking the value of 1.7956 and $\beta_3$ taking the value of 0.2708. The relationship between DBH and aboveground biomass $B_A$ for a given range is shown in Equation (7):

$$\begin{cases} B_{A1} = \alpha_4 D^{\beta_4} \\ B_{A2} = \alpha_5 D^{\beta_5} \end{cases}, \; 4.08 < D < 16.02 \tag{7}$$

where $B_{A1}$ represents the above-ground biomass including only the trunk and branch parts, with values of 0.0319 for $\alpha_4$ and 2.8303 for $\beta_4$; $B_{A2}$ represents the above-ground biomass with four parts: trunk, branch, leaf and deciduous leaf, and the value of $\alpha_5$ is 0.0673 and $\beta_5$ is 2.5651. The relationship between DBH and belowground biomass $B_U$ for a given range is shown in Equation (8):

$$B_U = \alpha_6 \ln D - \beta_6, \; 4.08 < D < 16.02 \tag{8}$$

where $B_U$ mainly includes two parts, root stump and thick root, and the value of $\alpha_6$ is taken as 3.3678 and the value of $\beta_6$ is taken as 4.0467.

It should be noted that not all data in practical measurements can be described by the fitting function, and some monitoring data may be distributed around the fitted curve. The goodness-of-fit $R^2$ is a statistic used to evaluate the degree of fit of the regression line to the sample observations. In the process of regression fitting, the value of the fitted object $y$ is mainly influenced by two factors: one is the different values of the explanatory variable $x$, and the other is the variation of random factors. Taking linear regression as an example, $R^2$ can be expressed as:

$$R^2 = 1 - \frac{\sum_{i=1}^{n} (y_i - z_i)^2}{\sum_{i=1}^{n} (y_i - \overline{y})^2} \tag{9}$$

$$\overline{y} = \frac{1}{n} \sum_{i=1}^{n} y_i \tag{10}$$

where $R^2$ represents the percentage of the regression sum of squares in the total sum of squares of variance for object $y$, $y_i$ is the sample observation, and $\overline{y}$ is the point on the regression line as shown in Equation (10). From Equation (9), we know that the range of

$R^2$ is (0, 1), and the better fit is when $R^2$ is closer to 1. For Equations (5)–(8), the $R^2$ ranges from 74% to 95% as can be seen from Table 2. When the amount of data $n \geq 100$, $R^2 \geq 87\%$. The physiological data obtained from real-time calculations based on DBH and tree growth equations are continuous. However, the tree model presents the same visual changes in tree height following the update cycle of Table 1, in which the measurement period of DBH still has an impact on it. In addition, other monitoring data, such as transpiration rate, stomatal conductance, and leaf water potential, which are related to stem sap flow, can also be simulated by this method.

**Table 2.** Goodness of fit ($R^2$) of tree growth equations.

| Physiological Indicators $y$ | Data Volume $n$ (#) | $R^2$ (%) |
|---|---|---|
| Tree height $H$ (m) | 1088 | 87.11% |
| Total biomass $B_{S1}$ (kg) | 100 | 92.70% |
| Total biomass $B_{S2}$ (kg) | 100 | 92.28% |
| Above-ground biomass $B_{A1}$ (kg) | 100 | 94.94% |
| Above-ground biomass $B_{A2}$ (kg) | 100 | 94.12% |
| Below-ground biomass $B_{U}$ (kg) | 85 | 74.32% |

## 3. Results

In this paper, we use Unity3D to initially build a digital twin-oriented poplar plantation system based on a virtual forest modeling and data analysis framework. It can run on a 64-bit PC equipped with Windows 10 operating system, Intel(R) Core(TM) i5-8250U CPU, and 8GB running memory.

### 3.1. Virtual Forest Results

In terms of virtual plantation forest, modeling of tree point clouds leads to a tree trunk model (Figure 9a). The simulation of different foliage densities using the transition particle flow method is shown in Table 3, which shows that the tree model has a stable emission ratio close to 44% at k=80%, and the storage size is still less than 1MB at an initial emission particle number of 15000. The first row of foliage particle data is selected into the trunk model to obtain the results (Figure 9b). In this paper, three different poplar tree models were selected to be integrated into the virtual plantation forest (Figure 9b–d).

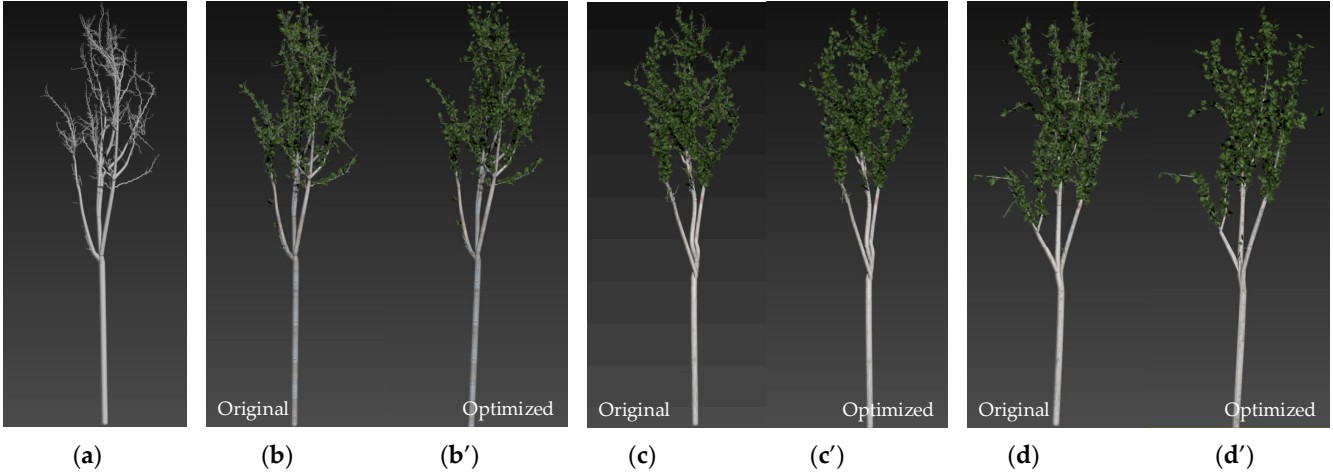

    (**a**)         (**b**)         (**b′**)         (**c**)         (**c′**)         (**d**)         (**d′**)

**Figure 9.** Models of *Triploid Populus Tomentosa*. (**a**) Tree stem model built using the AdTree method. (**b**) Tree model obtained by emitting leaf particles from the model in (**a**). (**b′**) Optimized model based on the model in (**b**). (**c**,**d**) Other models. (**c′**) Model optimized based on the model in (**c**). (**d′**) Model optimized based on the model in (**d**).

**Table 3.** Comparison of number of leaf particles and storage size.

| Initial Number of Particles Emitted (#) | Actual Number of Particles Emitted (#) | Particle Emission Ratio (%) | Storage Size (KB) |
|---|---|---|---|
| 15,000 | 6677 | 44.5 | 972 |
| 10,000 | 4443 | 44.43 | 689 |
| 8000 | 3546 | 44.3 | 553 |
| 4000 | 1773 | 44.3 | 285 |

Figure 9b′–d′ correspond to the results of the optimized tree models, respectively. The data comparison of the optimized tree models are shown in Table 4. The number of points, lines, and surfaces of the optimized model are almost 17% or less than the original number, and the storage size is reduced by more than 67%. As can be seen from Figure 9 and Table 4, the constructed tree model can maintain a certain degree of realism and can greatly reduce the size of the scene memory occupation. Other variables and factors such as soil, moisture, and climate in the plantation forest are not considered in this paper at this time. Adding models such as equipment modeled by 3dsMax, a more complete virtual poplar plantation forest is shown in Figure 10.

**Table 4.** Before and after optimization of the number of points, lines and surfaces of the tree model.

| Tree Models | Number of Vertices (#) | Number of Lines (#) | Number of Surfaces (#) | Storage Size (KB) | Frame Rate (FPS) |
|---|---|---|---|---|---|
| Figure 9b | 141,244 | 669,723 | 235,484 | 13,716 | 97.640 |
| Figure 9b′ | 23,777 | 67,116 | 22,372 | 4624 | 74.144 |
| Figure 9c | 140,844 | 667,826 | 226,672 | 13,029 | 83.96 |
| Figure 9c′ | 15,435 | 43,569 | 9521 | 385 | 75.76 |
| Figure 9d | 148,854 | 705,807 | 235,484 | 14,113 | 141.09 |
| Figure 9d′ | 10,777 | 30,420 | 6360 | 4206 | 46.02 |

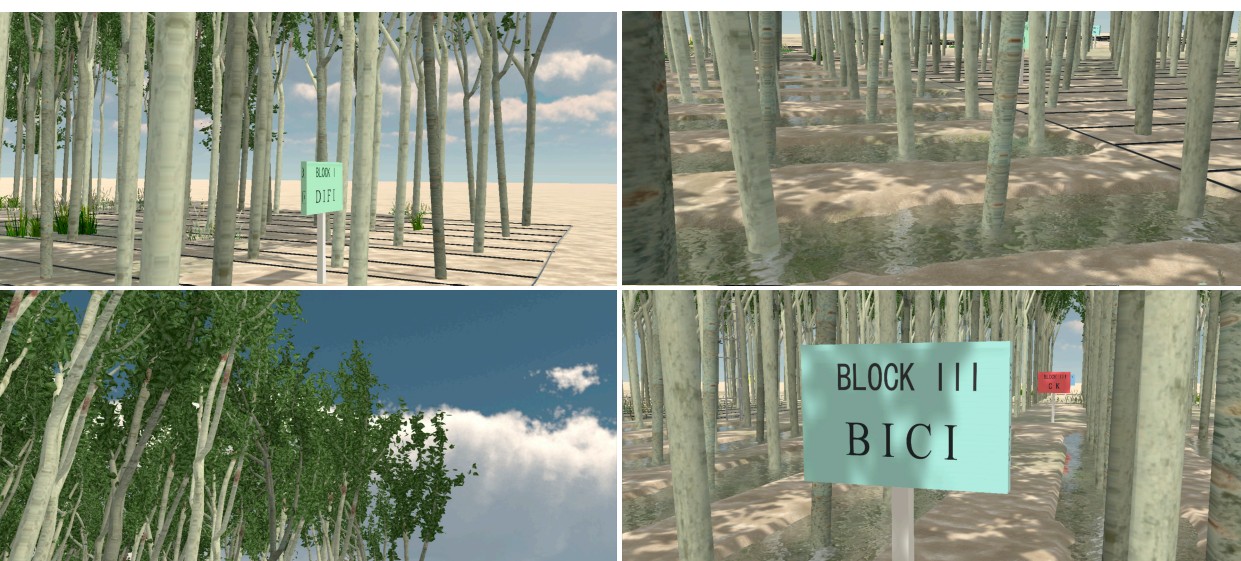

**Figure 10.** Virtual poplar plantation scene display.

### 3.2. Plantation Forest System

Unity3D's UGUI is used as a tool to realize the display of data and certain interface designs. The main interface of the system (Figure 11) is based on the resolution of 1280 × 720, mainly including the top information bar, the left and right task bars, and the bottom switch tour view function button. The left task bar currently displays the data information and trends of the three monitoring devices. The bottom of the right task bar is the monitoring

camera view of the poplar plantation weather station. The right taskbar is the main function implementation area, with three tabs for switching. The results of the "Tree diameter at breast height" and "Related predictions" functions are shown in Figure 12. Figure 12a shows mainly the DBH plot of trees with date, and Figure 12b–g shows the trend plot fitted according to the DBH. From them, we can visually observe that the trend of DBH and each data is basically consistent with the results fitted in Section 2.3.3. In addition, the tree growth data can be output by directly entering the tree DBH.

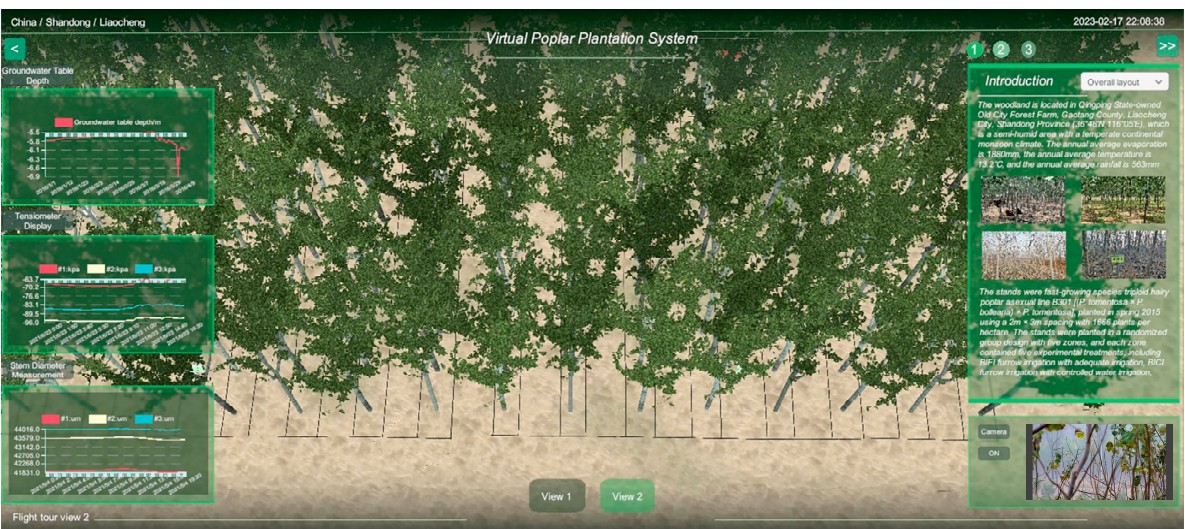

**Figure 11.** Virtual poplar plantation system's main interface display. The left and right sides are the task bars for the corresponding functions.

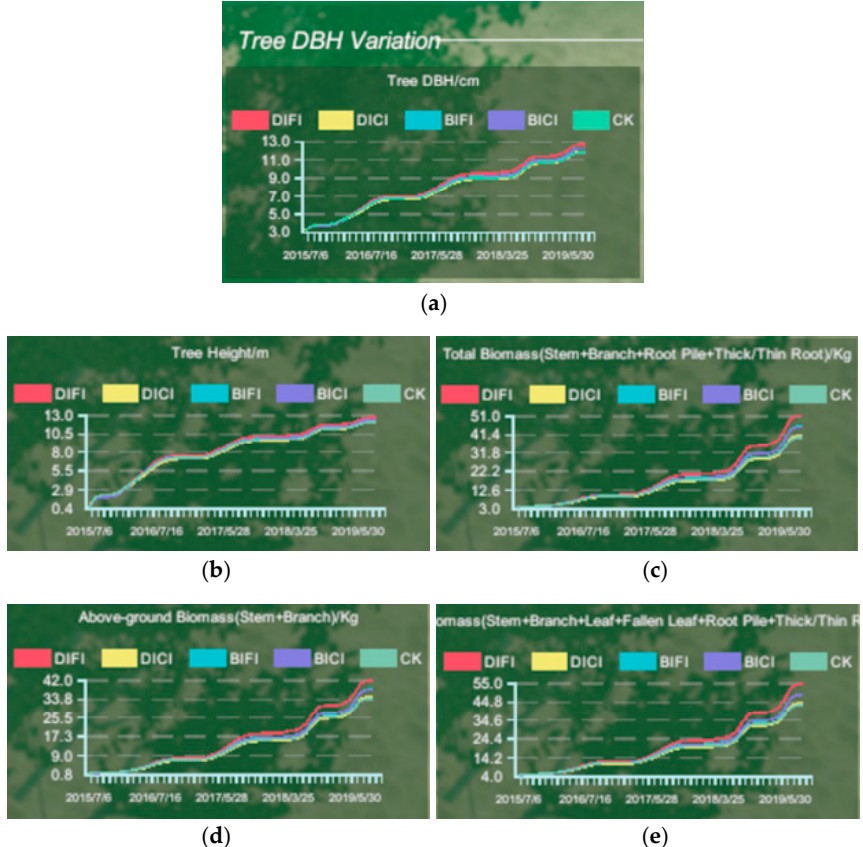

**Figure 12.** *Cont.*

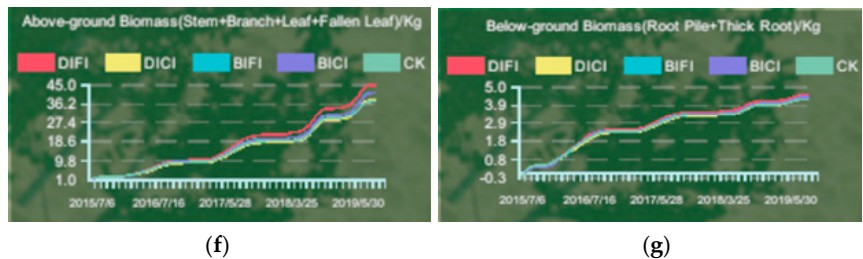

(**f**)                                                                    (**g**)

**Figure 12.** DBH-related trend plots of predicted stand status results. The five color icons represent the five different treatments (not discussed in this paper). (**a**) Trend chart of tree DBH. (**b**) Trend chart of tree height. (**c**) Trend chart of total biomass (excluding leaves and defoliation). (**d**) Trend chart of above-ground biomass (excluding leaves and defoliation). (**e**) Trend chart of total biomass. (**f**) Trend chart of above-ground biomass. (**g**) Trend chart of below-ground biomass.

## 4. Discussion

The architecture in this paper is developed based on three main dimensions: physical entities, virtual entities, and the connection between the two. Compared with the five-dimensional architecture of industry-oriented digital twin proposed by Tao et al. [46], although the two dimensions of data and services are missing, this paper's framework constitutes its own set of systems that are still feasible based on the results of the virtual plantation forest system for digital twin initially shown in Section 3.2. It is also possible to summarize the digital twin solution under multiple dimensions based on the five-dimensional model. The present study focuses on a poplar plantation as the main research object. Compared with the large-scale forest digital twin framework proposed by Buonocore et al. [47], the factors considered in this study are relatively few, but it is still expected to derive to larger-scale forest frameworks and target research and implementation through the study of this forest land. From the experimental results, the proposed framework in this paper can meet the eight evaluation criteria for digital twin models proposed by Zhang et al. [48], such as effectiveness, integrality, and intuitiveness. This framework performs analysis by establishing geometric and physical models, and the real data used in the forest land are both effective and standardized. However, the framework still lacks intelligent content such as model adaptability and self-learning ability. In addition, the established model only considers a single tree species, while there are multiple species and factors in actual forest land, and their interactions are very complex and uncertain, thus limiting the accuracy and applicability of the model. Therefore, for the optimization of the overall digital twin architecture, a more complete virtual forest digital twin framework can be constructed based on these evaluation criteria.

In this study, we have demonstrated the feasibility of the research framework by initially constructing a virtual poplar plantation system. Based on existing large-scale point cloud data, we reconstructed virtual forests using a single-tree modeling method, and the structure of the model was determined solely by the structure of the point cloud. Fourcaud et al. [49] discussed the importance of plant structure in growth models and their applications, and summarized the problems of related work. There have been works and studies on reconstruction of large-scale point cloud data [50], but there is still little research on the reconstruction of large-scale forest point clouds and the modeling and processes of simulating tree growth based on the real structure of plants. In addition, due to the interactions among multiple species and factors, there are still difficulties in conducting mixed modeling for multiple species. Therefore, in future research, it is necessary to consider incorporating plant growth models and more factors into virtual forests to achieve more realistic simulation effects. To achieve real-time data exchange and interoperability, reliance on intelligent devices is necessary. Currently, the framework relies on the periodic retrieval of data from the database by the system, lacking a real-time data transmission monitoring mechanism [51]. In future work, a fast update model based on real-time IoT communication protocols [52] can be considered to improve data transmission

performance. In addition, machine learning algorithms can be used to infer forest biomass for data analysis and derivative functions [53].

A virtual plantation forest system oriented towards digital twin can provide forestry researchers with a platform to simulate real forest environments, facilitating better experimentation and research. During the research process, the system can help researchers better understand and analyze data through feedback and visualization, and provide data sharing functions through data storage and management, making it more convenient and efficient to assist in adjusting forest management strategies. With future improvements to the system, researchers can better simulate the effects of different environmental factors on tree growth, further enhancing research and management of plantation forests. For the operation of the system, it is currently established to run on a computer with only a CPU, and the system is based on a C/S architecture [54].The C/S architecture allows for direct interface operation through the client and a more secure data connection method. The B/S architecture [55] can better meet the requirements of network interconnection and information sharing, and is more efficient and convenient for the management of the virtual forest. In the future, the system can be further improved and optimized by using a computer with GPU, and a system based on B/S architecture can be built. Considering the introduction of external VR devices for immersive interaction in the virtual forest can also be further explored for realizing the digital twin of forest land [56].

## 5. Conclusions

From the perspective of combining the forest and digital twin, this paper presents a framework of virtual forest modeling and data analysis. For digital forest management, a digital twin system architecture for a virtual poplar plantation is proposed. Based on the real data of *Triploid Populus Tomentosa* plantation, key problem analysis and function implementation are carried out for the two components of the framework, resulting in an initial and operational virtual forest system. For virtual forest modeling, the point cloud trunk model is constructed using the AdTree method, and the transition particle flow method is proposed to simulate the leaf particles. The resulting tree model can reduce memory pressure while maintaining realism. The virtual plantation forest scene is obtained by integrating and optimizing the tree model and other models. For data analysis, a SQL Server database is created for storage and then connected to the system. The monitoring data is pre-processed and parsed to obtain the tree growth equation by fitting, and then the tree diameter at breast height related data is analyzed and predicted. This enables the prediction and trend simulation of tree height and tree biomass data. The method is also applicable to other monitoring data of forest land. By utilizing virtual reality technology, it is possible to facilitate the overall construction of virtual forests from a three-dimensional perspective, and help achieve virtual forest management oriented towards digital twin.

**Author Contributions:** Conceptualization, W.L. and M.Y.; data curation, W.L., B.X. and Q.H.; formal analysis, W.L., M.Y. and B.X.; funding acquisition, M.Y. and B.X.; investigation, W.L. and M.Y.; methodology, W.L., M.Y. and Q.H.; project administration, W.L. and M.Y.; software, W.L.; supervision, M.Y.; validation, W.L.; visualization, W.L.; writing—original draft, W.L.; writing—review and editing, W.L., M.Y. and B.X. All authors have read and agreed to the published version of the manuscript.

**Funding:** This work was supported by the National Key Research and Development Program of China (No. 2021YFD2201203), the National Key Research and Development Program of China (No. 2019YFC1521104), the National Natural Science Foundation of China (No. 61402038).

**Data Availability Statement:** Not applicable.

**Acknowledgments:** Thank you to Liu (Yang Liu) and Li (Lingya Li) for providing us with knowledge and answers about forestry. Thank you to Fan (Guangpeng Fan) for providing us with answers about point cloud data modeling. Thank you to Zheng (Yili Zheng) and Zhu (Lichen Zhu) for providing us with surveillance camera videos of *Triploid Populus Tomentosa* plantation.

**Conflicts of Interest:** The authors declare no conflict of interest.

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
