# Peer review of "Framework of Virtual Plantation Forest Modeling and Data Analysis for Digital Twin"

_forests, doi:10.3390/f14040683_

Round 1

Reviewer 1 Report

The manuscript explores the feasibility for a virtual poplar plantation modeling, proposing a digital twin architecture. The paper shows how a virtual scene can be obtained based on LiDAR point cloud data and using a transition particle flow together with the AdTree method for tree modeling. Furthermore, a database of allometric functions fitting DBH data is incorporated.

The results therefore focus on memory optimization of the model, preserving the accuracy of forecasting tree growth equations. This is a preliminary study that illustrates the potential for using digital twin technology in the forestry field.

The paper lacks of a dedicated section on how this technology can help the user and the researcher to improve their activities, also highlighting the model limitations. For example, the topic of canopy space occupation and competition for light, which is a key factor in improving tree growth and biomass, is not addressed.

Then, the current model is able to work on a single tree species. How could a mix of other species be considered? What might be the limits in terms of memory and accuracy?

I would have expected more discussion on these points considering the aims of the journal and many studies in the forestry field on modelling tree growth.

Please see the specific comments to the text below.

Abstract

L14-5 The meaning of this sentence is unclear. What do you mean for “by using artificial methods”? How does a clear management purpose matter for biodiversity conservation? Please clarify

L15-7 The link with the previous sentence is missing. Please improve it.

Introduction

L37-9 Plantation forestry aims for timber production and thus has productive function more than to study forest regeneration

L83 Please state the LSTM acronym

L105 Please state the VR acronym

Methods

L221 Please describe the coefficient a

Results

Figure 12 Plots in this figure are too small. Please increase the size of each plot or consider to remove it if the content is already shown in the section 2.2.3

L496-500 Please rephrase these two sentences as the meaning is not clear

L500-7 I would suggest to reformulate this section because it is hard to follow your points

Reviewer 2 Report

The abstract is not clear and have very generic sentences such as line no 26, is not clear "The result shows that the framework is feasible."

The introduction seems too compact and need elaboration.

Authors needs to highlight clear objectives, contributions, and novelty compared to the state-of-the-art approaches?

Explanation of proposed frameworks needs details and mention why multiple frameworks has been used?

Vertices for the simplification of tree skeletons, justification for this approach is required.

Figure captions needs to be updated as these are too long and confusing.

Reference no 18, 35, 38, 39, 45, 46, 49, 53, 55, and 57 seem outdated. 

Overall paper organization needs to be revised.

Round 2

Reviewer 2 Report

Authors has polished the manuscript and may be published.